# Loss Decoupling for Task-Agnostic Continual Learning

**Yan-Shuo Liang and Wu-Jun Li**[*]
National Key Laboratory for Novel Software Technology,
Department of Computer Science and Technology, Nanjing University, P. R. China
liangys@smail.nju.edu.cn,liwujun@nju.edu.cn

## Abstract

Continual learning requires the model to learn multiple tasks in a sequential order. To perform continual learning, the model must possess the abilities to maintain performance on old tasks (stability) and adapt itself to learn new tasks (plasticity). Task-agnostic problem in continual learning is a challenging problem, in which task identities are not available in the inference stage and hence the model must learn to distinguish all the classes in all the tasks. In task-agnostic problem, the model needs to learn two new objectives for learning a new task, including distinguishing new classes from old classes and distinguishing between different new classes. For task-agnostic problem, replay-based methods are commonly used. These methods update the model with both saved old samples and new samples for continual learning. Most existing replay-based methods mix the two objectives in task-agnostic problem together, inhibiting the models from achieving a good trade-off between stability and plasticity. In this paper, we propose a simple yet effective method, called loss decoupling (LODE), for task-agnostic continual learning. LODE separates the two objectives for the new task by decoupling the loss of the new task. As a result, LODE can assign different weights for different objectives, which provides a way to obtain a better trade-off between stability and plasticity than those methods with coupled loss. Experiments show that LODE can outperform existing state-of-the-art replay-based methods on multiple continual learning datasets.

## 1 Introduction

Continual learning requires the model to learn multiple tasks in a sequential order [31]. However, neural network models suffer from a phenomenon called catastrophic forgetting (CF) [12], in which the performance of the network on the old tasks degrades significantly after it learns a new task. To address this challenge and enable continual learning, the model must possess the abilities to maintain performance on old tasks (stability) and adapt itself to learn new tasks (plasticity). Nevertheless, an excess of stability or plasticity can interfere with the other [39], and hence the model needs to make a trade-off between stability and plasticity [39, 31].

There exist two different kinds of problems, task-agnostic problem and task-aware problem, in continual learning. Task-agnostic problem [39] in continual learning is a challenging problem, in which task identities are not available in the inference stage and hence the model must learn to distinguish all the classes in all the tasks. In contrast, task-aware problem [28] in continual learning enables the model to get task identities in the inference stage. Therefore, the model for task-aware problem only needs to distinguish the classes belonging to the same task. The difference between task-agnostic problem and task-aware problem [28, 29] shows that the model in task-agnostic problem

---

[*]Wu-Jun Li is the corresponding author.

37th Conference on Neural Information Processing Systems (NeurIPS 2023).

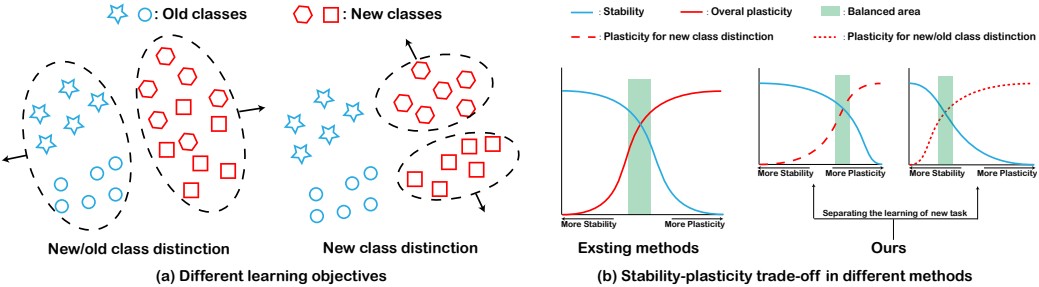

Figure 1: (a) shows two different learning objectives for learning the new task. The left side of (b) shows the trade-off between stability and plasticity in most existing methods, which mix different learning objectives. The right side of (b) shows that our method separates the learning of a new task into two objectives and hence separately considers the trade-off between stability and plasticity for these two learning objectives. y-axis in (b) represents the model's abilities, including plasticity and stability.

needs to learn two objectives for learning a new task, including distinguishing new classes from old classes (called new/old class distinction) and distinguishing between different new classes (called new class distinction). Figure 1 (a) illustrates these two learning objectives.

Many methods have been proposed for continual learning, including regularization-based methods [45, 20, 2], expansion-based methods [34, 18, 23] and replay-based methods [3, 10, 4]. For task-agnostic problem, replay-based methods are commonly used. These methods use a memory buffer to maintain a small portion of samples from the old classes. When learning a new task, the model retrieves old samples from memory and updates the parameters with both new and old samples. As illustrated in Figure 1 (b) from the methodological perspective, most existing continual learning methods [9, 7, 43] mix the two objectives in task-agnostic problem together (discussed later in Section 3.1). But these two learning objectives may cause different degrees of forgetting in continual learning and thus different trade-off strategies between stability and plasticity are required for these two learning objectives. More specifically, if a new learning objective leads to more forgetting, a good continual learner should pay more attention to the model's stability. On the contrary, if a new learning objective leads to less forgetting, a good continual learner should pay more attention to the model's plasticity for this objective. However, when the model mixes different learning objectives together, adjusting one of the learning objectives may influence others, inhibiting the model from achieving a good trade-off between stability and plasticity.

In this paper, we propose a simple yet effective method, called loss decoupling (LODE), for task-agnostic continual learning. The main contributions of this paper are listed as follows:

- By deeply analyzing the impacts of new/old class distinction and new class distinction, we find that these two learning objectives cause different degrees of forgetting. Therefore, mixing these two objectives together is detrimental for the model to make a good trade-off between stability and plasticity.

- LODE separates the two objectives for the new task by decoupling the loss of the new task. As a result, LODE can assign different weights for different objectives, which provides a way to obtain a better trade-off between stability and plasticity than those methods with coupled loss.

- Experiments show that LODE can outperform existing state-of-the-art replay-based methods on multiple continual learning datasets.

## 2 Related Work

Continual learning can be offline or online. In offline continual learning setting [17, 32, 44, 24], the model receives the entire dataset of a new task and updates its parameters multiple times over this dataset. In online continual learning setting [11, 4, 10], the data from each task is sequentially concatenated as a non-stationary data stream and each data of each task can only appear once in the

data stream. Hence, the model can only receive a mini-batch of samples from the data stream at a time and update its parameters based on this mini-batch.

Different types of methods have been proposed for continual learning, including regularization-based methods [45, 20, 2], expansion-based methods [34, 18, 23] and replay-based methods [3, 10, 4]. For task-agnostic problem, replay-based methods are commonly used. Some replay-based methods [40, 42] achieve replay by learning a generative model for generating old samples. However, learning a good generative model is challenging in some settings, like online continual learning [13]. Some replay-based methods like experience replay (ER) [9] maintain a memory buffer to save old samples and replay them with new samples. Some methods combine experience replay with knowledge distillation [15]. Specifically, these methods store either a old model [43] or the outputs of the old models [7]. Some methods [25] like experience replay with asymmetric cross entropy (ER-ACE) [8] improve ER by changing the loss of the new task so that the model can avoid large representation drift. Some methods like error-sensitive modulation experience replay (ERMER) [36] and complementary learning system experience replay (CLS-ER) [5] use an extra set of model's parameters to aggregate the knowledge of different tasks. Other methods try to search for valuable samples [3] or optimize the distribution of the memory [19, 41] to overcome forgetting. However, all these methods mix two new learning objectives in Figure 1 (a), inhibiting the models from achieving a good trade-off between stability and plasticity.

## 3 Methodology

In this section, we first formulate the problem in continual learning. Then, we deeply analyze the problem of mixing different learning objectives together in existing methods. After that, we propose our method called loss decoupling (LODE), which can be used in both offline and online continual learning settings. Finally, we discuss the relation between LODE and existing methods.

### 3.1 Problem Formulation

Continual learning requires the model to learn from multiple tasks in a sequential order. We use $\mathcal{D}_t = \{\boldsymbol{x}_i^t, y_i^t\}^{N_t}$ to denote the dataset of the $t$-th task, where $\boldsymbol{x}_i^t$ denotes an input sample and $y_i^t$ denotes its label. $N_t$ denotes the number of samples for task $t$. The total number of tasks is denoted by $T$. When the model learns a new task, the model can obtain limited or no data from the old tasks, potentially leading to catastrophic forgetting. In this work, we consider a challenging continual learning problem called task-agnostic problem, in which task identities are not available in the inference stage. In the task-agnostic problem, when the model learns on a new task $t$, some new classes are presented to the model. The model must possess the abilities to maintain performance on old classes (stability) and adapt itself to learn new classes (plasticity).

Replay-based methods [16, 7] maintain a memory buffer $\mathcal{M}$ with limited size to store a small portion of old samples. When receiving a mini-batch of new samples $\mathcal{B}_t$ from a new task $t$, the model retrieves a mini-batch of samples $\mathcal{B}_\mathcal{M}$ from $\mathcal{M}$ and replays them with the new samples $\mathcal{B}_t$ to achieve a trade-off between stability and plasticity. The losses used in most existing replay-based methods can be written as follows:

$$\mathcal{L} = \frac{1}{|\mathcal{B}_t|} \sum_{i=1}^{|\mathcal{B}_t|} \mathcal{L}_{new}(f_{\boldsymbol{\Theta}}(\boldsymbol{x}_i^t), y_i^t) + \frac{1}{|\mathcal{B}_\mathcal{M}|} \sum_{i=1}^{|\mathcal{B}_\mathcal{M}|} \mathcal{L}_{rep}(f_{\boldsymbol{\Theta}}(\boldsymbol{x}_i^\mathcal{M}), y_i^\mathcal{M}). \tag{1}$$

Here, $\mathcal{L}_{new}$ is the loss for the new task and is mainly for the plasticity of the model. In contrast, $\mathcal{L}_{rep}$ is the replay loss and is mainly for the stability of the model. We follow most existing works [7, 43] and assume that $\mathcal{L}_{new}$ is a cross-entropy loss. $\mathcal{L}_{rep}$ usually varies for different methods. For example, $\mathcal{L}_{rep}$ can be a cross-entropy loss [3, 9] or a combination of cross-entropy and regularization losses [7, 5, 36]. Based on Figure 1 (a), we can find that $\mathcal{L}_{new}$ is not only related to new/old class distinction, but also related to new class distinction in the task-agnostic problem. Hence, existing methods with the loss in (1) mix the two different learning objectives (new/old class distinction and new class distinction) together.

Note that there exists a type of replay-based methods [43, 17, 14, 38] directly sampling training data from $\mathcal{D}_t \cup \mathcal{M}$, which makes their losses can not be written in the form shown in (1). In some challenging continual learning settings like online continual learning setting, the model cannot access

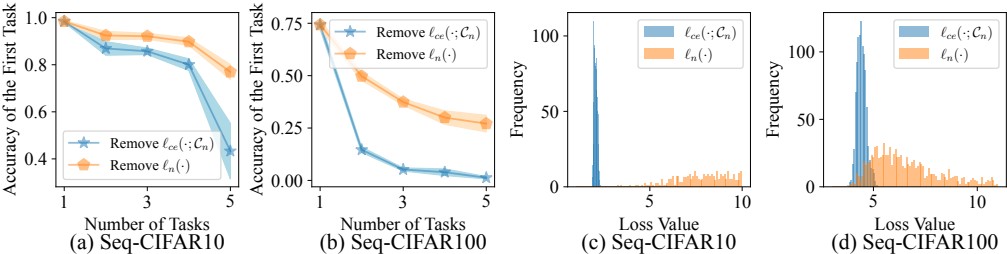

Figure 2: (a) and (b) show the variation of the first task's accuracy on different datasets. (c) and (d) show the distribution of different losses over different datasets before learning the second task. Here, the frequency corresponding to each loss value represents the number of samples with this loss value.

the entire dataset of a new task. At this time, it is impossible for the model to sample from $\mathcal{D}_t \cup \mathcal{M}$. On the contrary, loss in (1) can be adapted to online continual learning by sampling $\mathcal{B}_t$ from the data stream $\mathcal{S}$ and sampling $\mathcal{B}_\mathcal{M}$ from $\mathcal{M}$ separately. Hence, in this work, we focus on the methods whose loss function has the form in (1). In Section 4 and Appendix, we also compare our method with the methods that sample training data from $\mathcal{D}_t \cup \mathcal{M}$.

## 3.2    Analyzing Learning Objectives by Decoupling Loss

When learning the $t$-th task, we assume there are $m$ old classes, which are denoted as $\mathcal{C}_o = \{1, 2, ..., m\}$. We also assume there are $n$ new classes, which are denoted as $\mathcal{C}_n = \{m + 1, m + 2, ..., m + n\}$. Then, for a given new sample $(\boldsymbol{x}, y)$ where $y \in \mathcal{C}_n$, we can compute its loss $\mathcal{L}_{new}$ as

$$\mathcal{L}_{new}(f_\Theta(\boldsymbol{x}), y) = \ell_{ce}(f_\Theta(\boldsymbol{x}), y) = -\log\left(\frac{\exp(o_y)}{\sum_{i=1}^{m+n} \exp(o_i)}\right). \tag{2}$$

Here, $\ell_{ce}(\cdot)$ denotes the cross-entropy loss. $[o_1, o_2, ..., o_m, o_{m+1}, ..., o_{m+n}]$ denotes the logits outputed by $f_\Theta(\boldsymbol{x})$. We decouple the loss $\mathcal{L}_{new}(f_\Theta(\boldsymbol{x}), y)$ according to the two learning objectives in Figure 1 (a):

$$\mathcal{L}_{new}(f_\Theta(\boldsymbol{x}), y) = -\log\left(\frac{\exp(o_y)}{\sum_{i=m+1}^{m+n} \exp(o_i)}\right) - \log\left(\frac{\sum_{i=m+1}^{m+n} \exp(o_i)}{\sum_{i=1}^{m+n} \exp(o_i)}\right) \tag{3}$$

$$= \ell_{ce}(f_\Theta(\boldsymbol{x}), y; \mathcal{C}_n) + \ell_n(f_\Theta(\boldsymbol{x})).$$

Here, we use $\ell_{ce}(\cdot; \mathcal{C}_n)$ to denote the cross-entropy loss restricted to new classes $\mathcal{C}_n$. Obviously, $\ell_{ce}(f_\Theta(\boldsymbol{x}), y; \mathcal{C}_n)$ is related to new class distinction. $\ell_n(f_\Theta(\boldsymbol{x}))$ is related to new/old class distinction. Note that both $\ell_{ce}(f_\Theta(\boldsymbol{x}), y; \mathcal{C}_n)$ and $\ell_n(f_\Theta(\boldsymbol{x}))$ are for the plasticity of the model and may cause catastrophic forgetting. Furthermore, loss $\ell_{ce}(f_\Theta(\boldsymbol{x}), y; \mathcal{C}_n)$ and loss $\ell_n(f_\Theta(\boldsymbol{x}))$ have the same weight in (1) due to the coupling property.

We use experience replay (ER) [9], which is one of the most popular replay-based methods and can be expressed in the form of (1), to evaluate the impact of $\ell_{ce}(\cdot; \mathcal{C}_n)$ and $\ell_n(\cdot)$. Specifically, we first let the model learn on the first task through valina stochastic gradient descent [21]. Then, before learning the subsequent tasks, we remove one of the two losses in (3) from ER and analyze the forgetting of the first task. The experiments are conducted on two datasets Seq-CIFAR10 and Seq-CIFAR100, which will be introduced in Section 4. The experimental settings also follow the descriptions in Section 4.1. Figure 2 (a) and Figure 2 (b) show the accuracy of the first task when the model learns subsequent tasks. Here, 'Remove $\ell_{ce}(\cdot; \mathcal{C}_n)$' means that we remove $\ell_{ce}(\cdot; \mathcal{C}_n)$ from (1) after learning the first task. 'Remove $\ell_n(\cdot)$' means that we remove $\ell_n(\cdot)$ from (1) after learning the first task. The results show that removing $\ell_n(\cdot)$ results in less forgetting of the first task than removing $\ell_{ce}(\cdot; \mathcal{C}_n)$. In other words, $\ell_n(\cdot)$ leads to more forgetting than $\ell_{ce}(\cdot; \mathcal{C}_n)$.

It is intuitively reasonable that $\ell_n(\cdot)$ leads to more forgetting than $\ell_{ce}(\cdot; \mathcal{C}_n)$. First, since replay-based methods usually keep limited samples in memory, when the model learns a new task, it has access to much fewer samples from the old classes than from the new classes. Therefore, utilizing loss $\ell_n(\cdot)$ to learn to distinguish between new classes and old classes introduces a risk of biasing the model towards the new classes, potentially leading to serious catastrophic forgetting. In contrast, loss

$\ell_{ce}(\cdot; \mathcal{C}_n)$ is independent of the old classes, thereby avoiding introducing a risk of biasing the model towards the new classes. Second, in Figure 2 (c) and Figure 2 (d), we also show the value of the losses $\ell_{ce}(\cdot; \mathcal{C}_n)$ and $\ell_n(\cdot)$ before using ER to learn task 2. We can find that the value of $\ell_n(\cdot)$ is larger than the value of $\ell_{ce}(\cdot; \mathcal{C}_n)$ on both Seq-CIFAR10 and Seq-CIFAR100. This finding is consistent with existing works [8, 36], which suggest that larger losses for the new task may result in larger feature drift, leading to more forgetting. In Appendix, we also provide the TSNE visualizations when either $\ell_n(\cdot)$ or $\ell_{ce}(\cdot; \mathcal{C}_n)$ is removed, further confirming that $\ell_n(\cdot)$ leads to more forgetting than $\ell_{ce}(\cdot; \mathcal{C}_n)$.

Since $\ell_n(\cdot)$ leads to more forgetting than $\ell_{ce}(\cdot; \mathcal{C}_n)$, treating these two losses equally is not reasonable. In particular, based on the above analysis, we can find that a good continual learner should assign a larger weight to $\ell_{ce}(\cdot; \mathcal{C}_n)$ and a smaller weight to $\ell_n(\cdot)$. However, loss in (1) fails to achieve this goal due to the coupling property.

### 3.3 Loss Decoupling for Continual Learning

Section 3.2 has demonstrated the impact of different learning objectives on the model's forgetting and the issue of the coupling property in existing replay-based methods. To address this issue, we propose a new method called loss decoupling (LODE), which removes the coupling property present in existing methods.

Specifically, our LODE uses the following loss to perform continual learning:

$$\mathcal{L} = \frac{1}{|\mathcal{B}_t|} \sum_{i=1}^{|\mathcal{B}_t|} \left( \beta_1 \ell_{ce}(f_\Theta(\boldsymbol{x}_i^t), y_i^t; \mathcal{C}_n) + \beta_2 \ell_n(f_\Theta(\boldsymbol{x}_i^t), y_i^t) \right) + \frac{1}{|\mathcal{B}_\mathcal{M}|} \sum_{i=1}^{|\mathcal{B}_\mathcal{M}|} \mathcal{L}_{rep}(f_\Theta(\boldsymbol{x}_i^\mathcal{M}), y_i^\mathcal{M}).$$

(4)

Here, $\beta_1$ and $\beta_2$ are two coefficients that control the weight of the two different learning objectives. The finding in Section 3.2 shows that LODE should set $\beta_2$ to be smaller than $\beta_1$, to make the model achieve a better trade-off between stability and plasticity than existing methods usually with coupled loss. Furthermore, since $\ell_n(\cdot)$ is for new/old class distinction, LODE sets $\beta_2$ proportional to the ratio $\frac{|\mathcal{C}_n|}{|\mathcal{C}_o|}$ to make the model not bias toward old or new classes. In contrast, since $\ell_{ce}(\cdot; \mathcal{C}_n)$ is only related to the new classes, LODE sets $\beta_1$ to be a constant value. More specifically, LODE sets $\beta_1$ and $\beta_2$ as

$$\beta_1 = C, \quad \beta_2 = \rho \frac{|\mathcal{C}_n|}{|\mathcal{C}_o|}.$$

(5)

Here, $C$ and $\rho$ are two hyperparameters. Note that when the number of tasks increases, the number of old classes also increases. In particular, when the number of old tasks is large, the number of old classes $|\mathcal{C}_o|$ is usually much larger than the number of new classes $|\mathcal{C}_n|$. At this time, $\beta_2$ is much smaller than $\beta_1$. Setting $\beta_2$ to be as large as $\beta_1$, or setting $\beta_1$ to be as small as $\beta_2$ fails to make the model achieve a good trade-off between stability and plasticity, which will be verified in the experiment.

Note that we do not specify the form of $\mathcal{L}_{rep}$ in (4). Therefore, our method can be combined with many replay-based methods with form (1) and improve these methods. Here, we give some examples which combine LODE with different state-of-the-art continual learning methods.

**Combining LODE with ER and DER++** Both experience replay (ER) [9] and dark experience replay++ (DER++) [7] can be written in the form of (1). Therefore, the combinations of our LODE with these two methods are direct. Specifically, given a new batch of samples $\mathcal{B}_t$, the model compute $\ell_{ce}(\cdot; \mathcal{C}_n)$ and $\ell_n(\cdot)$ according to (3). Then, the model computes $\mathcal{L}_{rep}$ with the old samples $\mathcal{B}_\mathcal{M}$ through the specific formulation in ER or DER++. Finally, the model can get the loss (4).

**Combining LODE with ESMER** A recent method called error sensitive modulation experience replay (ESMER) [36] suggests that the model should learn more from smaller losses to avoid large feature drift. The loss in ESMER is slightly different from the loss in (1). Specifically, ESMER assigns different weights to different new samples in $\mathcal{B}_t$ according to their loss values. Although the loss in ESMER is slightly different from the loss in (1), it still mixes the two different learning objectives together. Therefore, we can combine ESMER with our LODE through a similar form

---

**Algorithm 1** Loss Decoupling (LODE) for Continual Learning

1: **Input:** a sequence of tasks with datasets $\{\mathcal{D}_1, ..., \mathcal{D}_T\}$, a neural network model $f_\Theta(\cdot)$.
2: **Output:** a learned neural network model $f_\Theta(\cdot)$.
3: **while** Get a mini-batch of samples $\mathcal{B}_t$ from a task $t$ **do**
4:     Sample a mini-batch $\mathcal{B}_\mathcal{M}$ from memory $\mathcal{M}$;
5:     Specify the weights for the two different learning objectives by (5);
6:     Get the losses for learning objective through (3);
7:     Compute the final loss through (4).
8:     Perform backward propagation and update the model $f_\Theta(\cdot)$ through SGD;
9:     Update memory $\mathcal{M}$ with $\mathcal{B}_t$ through some memory update methods;
10: **end while**

---

to (4). The decoupled loss for ESMER can be written as follows:

$$\mathcal{L} = \frac{1}{|\mathcal{B}_t|} \sum_{i=1}^{|\mathcal{B}_t|} w_i \left( \beta_1 \ell_{ce}(f_\Theta(\boldsymbol{x}_i^t), y_i^t; \mathcal{C}_n) + \beta_2 \ell_n(f_\Theta(\boldsymbol{x}_i^t), y_i^t) \right) + \frac{1}{|\mathcal{B}_\mathcal{M}|} \sum_{i=1}^{|\mathcal{B}_\mathcal{M}|} \mathcal{L}_{rep}(f_\Theta(\boldsymbol{x}_i^\mathcal{M}), y_i^\mathcal{M}).$$

$$(6)$$

Here, $w_i$ is the weight assigned to new sample $i$ in ESMER. We can find that the loss in (6) not only modulates weights for different new samples but also modulates weights for the two different learning objectives.

Algorithm 1 gives the whole learning process of LODE.

### 3.4 Relation with Existing Methods

The loss in many existing methods has the coupling property like the loss in (1). However, some methods, from the perspective of this work, use the losses that show a certain degree of decoupling property. For example, the loss in ER-ACE [8] can be written as

$$\mathcal{L} = \frac{1}{|\mathcal{B}_t|} \sum_{i=1}^{|\mathcal{B}_t|} \mathcal{L}_{ce}(f_\Theta(\boldsymbol{x}_i^t), y_i^t; \mathcal{C}_n) + \frac{1}{|\mathcal{B}_\mathcal{M}|} \sum_{i=1}^{|\mathcal{B}_\mathcal{M}|} \mathcal{L}_{rep}(f_\Theta(\boldsymbol{x}_i^\mathcal{M}), y_i^\mathcal{M}). \qquad (7)$$

It can be seen from (7) that this method removes the term $\ell_n(\cdot)$ from (3) and only retains $\ell_{ce}(\cdot; \mathcal{C}_n)$. In particular, loss in (7) is a special case of the loss in (4). More specifically, we can get the loss in (7) by setting $\beta_1 = 1$ and $\beta_2 = 0$ in (4). SSIL [1] has a similar form to that in (7) but uses a different $\mathcal{L}_{rep}$ compared to ER-ACE. Due to the lack of $\ell_n(\cdot)$, the loss in (7) can only leverage the new classes' samples kept in memory to learn the objective of new/old class distinction. In experiments, we will show that setting $\beta_2 = 0$ in (4) performs worse than setting $\beta_2 \neq 0$ in (4).

Some existing methods also incorporate the idea of separating objectives in continual learning. However, these methods are primarily designed for task-aware problem. For instance, bilevel memory system with knowledge projection (BMKP) [37] requires the task identities to choose the corresponding knowledge representations during testing, making it unsuitable in the task-agnostic problem. Space decoupling (SD) [46] does not explicitly mention that it only considers task-aware problem, but its experiments completely follow some task-aware methods [35, 26], indicating its focus on task-aware problem.

## 4 Experiments

### 4.1 Experimental Settings

**Datasets** We use three popular datasets for evaluation, including Seq-CIFAR10 [3], Seq-CIFAR100 [10], and Seq-TinyImageNet [22]. Seq-CIFAR10 consists of 5 disjoint tasks with each task having 2 classes and 10k training samples. Seq-CIFAR100 consists of 5 disjoint tasks with each task having 20 classes and 10k training samples. Seq-TinyImageNet consists of 10 disjoint tasks with each task having 20 classes and 10k training samples. The statistics of different datasets are given in Appendix. All the experiments are for the task-agnostic problem.

Table 1: Classification results which are averaged across 5 runs.

| Keeping Extra Model | | Seq-CIFAR10 | | Seq-CIFAR100 | | Seq-TinyImageNet | |
|---|---|---|---|---|---|---|---|
| no | *joint* | 91.86±0.26 | | 70.10±0.60 | | 59.82±0.31 | |
| | *finetune* | 19.65±0.03 | | 17.41±0.09 | | 8.13±0.04 | |
| | **Buffer Size** | 500 | 5120 | 500 | 5120 | 500 | 5120 |
| no | SCR [27] | 57.95±1.57 | 82.47±0.44 | 23.06±0.22 | 45.02±0.67 | 8.37±0.26 | 18.20±0.48 |
| | PCR [25] | 65.74±3.29 | 82.58±0.42 | 28.38±0.46 | 52.51±1.61 | 11.88±1.61 | 26.39±1.64 |
| | MIR [3] | 63.93±0.39 | 83.73±0.97 | 27.80±0.52 | 53.73±0.82 | 11.22±0.43 | 30.60±0.40 |
| | ER-ACE [8] | 68.45±1.78 | 83.49±0.40 | 40.67±0.06 | 58.56±0.91 | 17.73±0.56 | 37.99±0.17 |
| | ER [9] | 61.78±0.72 | 83.64±0.95 | 27.69±0.58 | 53.86±0.57 | 10.36±0.11 | 27.54±0.30 |
| | LODE (ER) | 68.87±0.71 | 83.73±0.48 | 41.52±1.22 | 58.59±0.48 | 17.77±1.03 | 38.34±0.04 |
| | DER++ [7] | 73.29±0.96 | 85.66±0.14 | 42.08±1.71 | 62.73±0.58 | 19.28±0.61 | 39.72±0.47 |
| | LODE (DER++) | **75.45±0.90** | **85.78±0.40** | **46.31±1.01** | **64.00±0.48** | **21.15±0.68** | **40.31±0.03** |
| yes | CLS-ER [5] | 70.73±0.54 | **85.73±0.29** | 51.21±0.84 | 60.17±0.38 | 29.44±1.66 | 45.66±0.47 |
| | TAMiL [6] | 74.25±0.31 | 84.82±1.77 | 50.62±0.23 | 63.77±0.43 | 27.83±0.41 | 43.00±0.56 |
| | iCaRL [33] | 61.60±2.03 | 72.01±0.62 | 49.59±0.95 | 54.23±0.28 | 20.01±0.50 | 30.34±0.18 |
| | BIC [43] | 52.63±2.46 | 79.98±1.49 | 37.06±0.60 | 60.43±0.61 | 29.82±0.88 | 37.60±0.23 |
| | SSIL [1] | 64.31±0.89 | 71.72±1.47 | 41.61±0.37 | 57.53±0.52 | 16.80±0.71 | 40.06±0.58 |
| | ESMER [36] | 71.48±0.98 | 79.19±0.68 | 52.37±0.87 | 63.99±0.13 | 30.97±1.12 | 44.07±0.52 |
| | LODE (ESMER) | **74.53±0.95** | 85.34±0.41 | **55.06±0.35** | **65.69±0.33** | **32.15±0.17** | **46.40±0.46** |

**Baselines** We compare our method with many state-of-the-art replay-based continual learning methods, including incremental classifier and representation learning (iCaRL) [33], bias correction (BIC) [43], separated softmax for incremental learning (SSIL) [1], experience replay (ER) [9], maximally interfere retrieval (MIR) [3], dark experience replay++ (DER++) [7], supervised contrastive replay (SCR) [27], proxy-based contrastive replay (PCR) [25], experience replay with asymmetric cross entropy (ER-ACE) [8], error sensitive modulating experience replay (ESMER) [36], complementary learning system experience replay (CLS-ER) [5], and task-specific attention modules in lifelong learning (TAMiL) [6]. For CLS-ER, we follow the existing method [6] and implement it with a single exponential moving average model. We also include two methods without continual learning, *joint* and *finetune*, in the comparison. Here, *joint* denotes the method which learns all the tasks jointly while *finetune* denotes the method which learns all the tasks sequentially without any memory. The accuracy of *joint* can be treated as the accuracy upper-bound and the accuracy of *finetune* can be treated as the accuracy lower-bound. Among the methods we mentioned above, some methods only maintain a single learning model to perform continual learning, while others require an extra model in memory for knowledge integration or distillation. Since keeping more models requires more memory, and memory cost is an important metric in continual learning [30], we group different methods by whether keeping extra model (refer to Table 1) to make a more fair comparison.

**Architecture and Training Details** We follow existing continual learning works [7, 6] and use standard ResNet18 as the neural network architecture in all the experiments unless otherwise stated. The experiments are built on top of the mammoth [7] continual learning repository in PyTorch like many existing works [7, 6]. We use stochastic gradient descent (SGD) to optimize the parameters. The batch size and replay size are set to 32 to follow the existing continual learning works [7, 6]. We also follow existing methods [7, 5] to set memory as 500 and 5120 for all the datasets. The hyperparameters are selected through a small validation set. For the experiments of all the methods on all the datasets, we apply random crops and horizontal flips to both newly coming samples and buffered (saved) samples like existing works [7, 5]. For each of our experiments, we report the average and standard deviation of the mean test accuracy of all the tasks across 5 runs with different seeds. More details of training and hyperparameters for different methods are given in Appendix.

## 4.2 Experimental Results

### 4.2.1 Accuracy

Table 1 shows the results of different methods on different datasets. Here, 'Keeping Extra Model' represents whether a method needs to keep an extra model for continual learning, as described

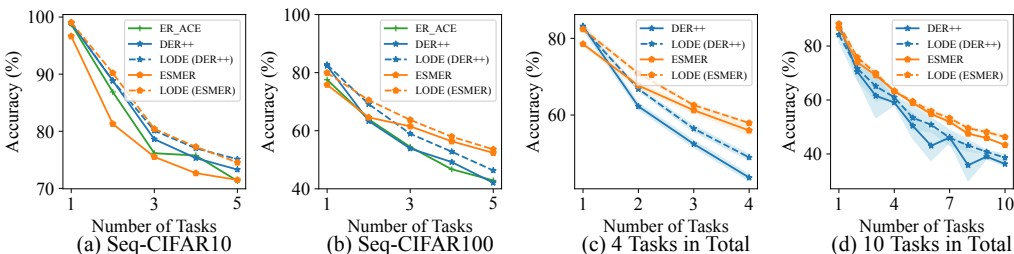

Figure 3: (a) and (b) show the variation of the accuracy for different methods on Seq-CIFAR10 and Seq-CIFAR100. (c) and (d) show the variation of accuracy on Seq-CIFAR100 with different number of tasks.

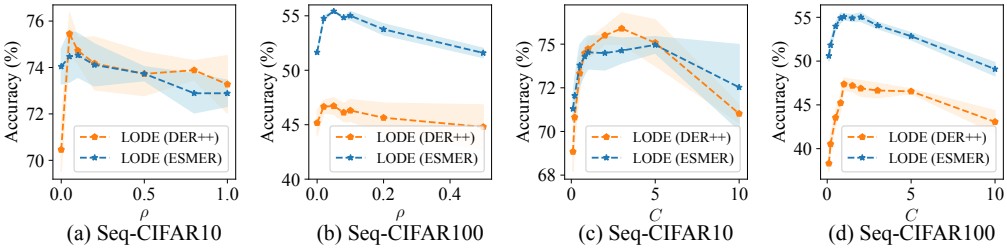

Figure 4: (a) and (b) show the variation of the accuracy for different $\rho$. (c) and (d) show the variation of the accuracy for different $C$.

in Section 4.1. We combine our LODE with ER, DER++ and ESMER, which can be written in the form of (1). We use LODE (ER), LODE (DER++) and LODE (ESMER) to denote them, respectively. Our experimental results confirm that the integration of our LODE method improves the performance of each combined method. For example, when compared to DER++ with buffer size 500, LODE (DER++) exhibits a $2.16\%$ improvement on Seq-CIFAR10, a $4.23\%$ improvement on Seq-CIFAR100 and a $1.87\%$ improvement on Seq-TinyImageNet. Notably, LODE (DER++) achieves the best performance on Seq-CIFAR10. LODE (ESMER) achieves the best performance on Seq-CIFAR100 and Seq-TinyImageNet.

Figure 3 (a) and Figure 3 (b) show the variation of accuracy on Seq-CIFAR10 and Seq-CIFAR100 after the learning of each task. As we can see, LODE improves DER++ and ESMER at the end of each task. Figure 3 (c) and Figure 3 (d) show the results on Seq-CIFAR100 with different numbers of tasks. When there are 4 tasks in Seq-CIFAR100, each task consists of 25 exclusive classes. Similarly, when there are 10 tasks in Seq-CIFAR100, each task consists of 10 exclusive classes. As we can see, when the number of tasks varies, LODE still gives improvements on different methods. In Appendix, we show the results of more methods on Seq-CIFAR100 with different numbers of tasks.

#### 4.2.2 Hyperparameter Analysis

We analyze the hyperparameters in LODE. We choose LODE (DER++) and LODE (ESMER) to analyze hyperparameters, as they represent the best performance of methods that retain or do not retain an extra model for continual learning, respectively.

We first vary the value of $\rho$ in (5) to show its impact on the performance of the model. Figure 4 (a) and Figure 4 (b) give the analysis on Seq-CIFAR10 and Seq-CIFAR100. Note that when $\rho = 0$, $\beta_2 = 0$ and the weight of $\ell_n(\cdot)$ in (4) is always zero. At this time, the loss in (4) degenerates to the loss in (7) and the model fails to get satisfactory performance. When the value of $\rho$ increases, $\beta_2$ also increases and the performance of the model first increases and then decreases. This phenomenon is intuitively reasonable since a larger weight for $\ell_n(\cdot)$ leads to more forgetting and thus influences the overall model performance.

We also vary the value of $C$ in (5) to show its impact on the performance of the model. Figure 4 (c) and Figure 4 (d) show the results of varying $C$ on Seq-CIFAR10 and Seq-CIFAR100, respectively. We can find that the performance of the model decreases significantly when $C$ is close to zero. In

Table 2: Ablation study on Seq-CIFAR10 and Seq-CIFAR100.

| | LODE (DER++) | | LODE (ESMER) | |
|---|---|---|---|---|
| | Seq-CIFAR10 | Seq-CIFAR100 | Seq-CIFAR10 | Seq-CIFAR100 |
| $\beta_1 = C, \beta_2 = \rho\frac{|\mathcal{C}_n|}{|\mathcal{C}_o|}$ (Ours) | $\mathbf{75.45}_{\pm 0.90}$ | $\mathbf{46.31}_{\pm 1.01}$ | $\mathbf{74.53}_{\pm 0.95}$ | $\mathbf{55.06}_{\pm 0.35}$ |
| $\beta_1 = \beta_2 = \rho\frac{|\mathcal{C}_n|}{|\mathcal{C}_o|}$ | $71.18_{\pm 0.80}$ | $37.49_{\pm 1.79}$ | $73.41_{\pm 0.40}$ | $45.64_{\pm 0.87}$ |
| $\beta_1 = \beta_2 = C$ | $73.80_{\pm 0.72}$ | $42.08_{\pm 1.71}$ | $73.08_{\pm 0.81}$ | $52.37_{\pm 0.87}$ |
| $\beta_1 = \rho\frac{|\mathcal{C}_n|}{|\mathcal{C}_o|}, \beta_2 = C$ | $73.19_{\pm 0.15}$ | $40.79_{\pm 0.12}$ | $72.38_{\pm 0.24}$ | $51.86_{\pm 0.35}$ |

Table 3: Classification results which are averaged across 5 runs in the online continual learning setting.

| Keeping Extra Model | | Seq-CIFAR10 | Seq-CIFAR100 | Seq-TinyImageNet |
|---|---|---|---|---|
| | SCR [27] | $69.49_{\pm 3.02}$ | $36.09_{\pm 0.82}$ | $20.04_{\pm 1.24}$ |
| | PCR [25] | $73.28_{\pm 1.83}$ | $34.89_{\pm 0.67}$ | $23.84_{\pm 0.60}$ |
| | ER-ACE [8] | $69.17_{\pm 1.64}$ | $35.24_{\pm 0.51}$ | $23.42_{\pm 0.34}$ |
| | MIR [3] | $71.10_{\pm 1.59}$ | $35.08_{\pm 1.32}$ | $20.64_{\pm 1.17}$ |
| no | ER [9] | $67.93_{\pm 2.04}$ | $34.40_{\pm 1.13}$ | $21.14_{\pm 0.72}$ |
| | LODE (ER) | $69.63_{\pm 1.41}$ | $36.91_{\pm 1.38}$ | $24.31_{\pm 0.82}$ |
| | DER++ [7] | $72.30_{\pm 0.99}$ | $34.72_{\pm 1.51}$ | $20.40_{\pm 1.02}$ |
| | LODE (DER++) | $\mathbf{74.00}_{\pm 0.08}$ | $\mathbf{37.82}_{\pm 1.16}$ | $\mathbf{25.30}_{\pm 1.80}$ |

particular, the model gets the best performance when the value of $C$ is between 1 and 5. Since $\rho$ and $C$ are highly different when the model achieves the best performance, decoupling the loss through (3) is necessary for continual learning.

### 4.2.3 Ablation Study

We change the value of $\beta_1$ and $\beta_2$ to show the effectiveness of setting $\beta_1$ and $\beta_2$ through (5). We first set the value of $\beta_1 = \beta_2$ in (4) to remove the decoupling property. There are two possibilities to set $\beta_1 = \beta_2$. The first possibility is to set $\beta_1 = \beta_2 = C$ and the second possibility is to set $\beta_1 = \beta_2 = \rho\frac{|\mathcal{C}_n|}{|\mathcal{C}_o|}$. Table 2 shows the results of these two possibilities, which are significantly inferior to our method. This means that separating the two different objectives by decoupling the loss of the new task is necessary for the model to achieve good performance. In Table 2, we also show the result of a variant by exchanging the value of $\beta_1$ and $\beta_2$, which means $\beta_1 = \rho\frac{|\mathcal{C}_n|}{|\mathcal{C}_o|}$ and $\beta_2 = C$. We can find that the performance of this variant is still significantly inferior to our method.

### 4.2.4 Online Continual Learning

We also perform experiment for the online continual learning setting [3, 10] where the datasets of different tasks are concatenated to a non-stationary data stream. Since online continual learning is more challenging than offline continual learning, existing methods [11] usually use larger memory in online continual learning, especially for challenging datasets such as Seq-CIFAR100 and Seq-TinyImageNet. Hence, we set the memory size to 5120 for all the datasets. To follow existing online continual learning methods [3, 10], the experimental settings we use here are different from those introduced in Section 4.1. These settings are introduced in Appendix.

Table 3 shows the results of different methods. We exclude those methods that have been implemented only in offline continual learning or those that have demonstrated lower performance in online continual learning in previous works [8, 25]. Similar to that in the offline continual learning setting, we can find that the integration of our LODE method also significantly improves the overall accuracy of each combined method. For example, when compared with DER++, LODE (DER++) exhibits a 1.70% improvement on Seq-CIFAR10, a 3.1% improvement on Seq-CIFAR100 and a 4.9% improvement on Seq-TinyImageNet.

## 5 Conclusion

In this work, we propose a new method called loss decoupling (LODE) for continual learning. Different from most existing replay-based methods which mixes the two different learning objectives together to learn the new task, LODE separates the two learning objectives for the new task by decoupling the loss of the new task. Experiments show that LODE can achieve a better trade-off between stability and plasticity than other baselines, and thus outperform other state-of-the-art replay-based methods across multiple datasets. Future work will extend LODE to other continual learning problem like task-aware problem and study the effectiveness of LODE for other types of tasks.

## Acknowledgment

This work is supported by NSFC Project (No.62192783), National Key R&D Program of China (No.2020YFA0713901), and Fundamental Research Funds for the Central Universities (No.020214380108).

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

## A    Statistics of Different Continual Learning Datasets

We give detailed statistics on datasets in this section. Specifically, Table 4 shows the detailed information of three datasets, including Seq-CIFAR10, Seq-CIFAR100, and Seq-TinyImageNet. These datasets are constructed by one of the following datasets: CIFAR10, CIFAR100 or TinyImageNet. For each task, $5\%$ of the training samples are divided into a validation set.

Table 4: Statistics on three datasets

|  | Seq-CIFAR10 | Seq-CIFAR100 | Seq-TinyImageNet |
|---|---|---|---|
| Task Number | 5 | 5 | 10 |
| Input Size | $3 \times 32 \times 32$ | $3 \times 32 \times 32$ | $3 \times 64 \times 64$ |
| Classes per Task | 5 | 20 | 20 |
| Training Samples | 10000 | 10000 | 10000 |
| Testing Samples | 1000 | 1000 | 1000 |

## B    More Experimental Details

### B.1    Hyperparameters

Table 5 provides the hyperparameters for different methods, which will be used for comparison. $\rho$ is selected among four different values $[0.01, 0.05, 0.1, 0.2]$ through the validation sets we split from the training sets. We do not carefully select hyperparameter $C$ but set it as 1 for all the experiments to reduce the reliance of the model on hyperparameters.

It is important to note that both DER++ and ESMER have their own specific hyperparameters that may affect the performance of the model. To ensure a fair comparison, we maintain the specific hyperparameters of each method consistent when combined with our LODE, except for the exponential decay hyperparameter $\alpha$ in ESMER. This means that we use the same hyperparameters settings as the original implementation when evaluating our combined models. For the exponential decay hyperparameter $\alpha$ in ESMER, we find that setting it to the same value as ESMER (0.999) makes the model too conservative to learn the new task in LODE (ESMER). Therefore, we set it to 0.998 in order to make the model not so conservative. Furthermore, we follow the mammoth continual learning repository [7] to set the specific hyperparameters in each method or get them by the validation sets we split from the training sets.

Table 5: List of hyperparameters for different methods

| Methods | Buffer | Split-CIFAR10 | | Split-CIFAR100 | | Split-TinyImageNet | |
|---|---|---|---|---|---|---|---|
|  |  | $C$ | $\rho$ | $C$ | $\rho$ | $C$ | $\rho$ |
| LODE (ER) | 500 | 1 | 0.1 | 1 | 0.01 | 1 | 0.05 |
|  | 5120 | 1 | 0.1 | 1 | 0.05 | 1 | 0.05 |
| LODE (DER++) | 500 | 1 | 0.05 | 1 | 0.1 | 1 | 0.1 |
|  | 5120 | 1 | 0.1 | 1 | 0.2 | 1 | 0.1 |
| LODE (ESMER) | 500 | 1 | 0.1 | 1 | 0.05 | 1 | 0.05 |
|  | 5120 | 1 | 0.1 | 1 | 0.2 | 1 | 0.1 |

The learning rate is determined by a grid-search. Specifically, the open-source code of the existing work DER [7] has conducted a grid-search to determine hyperparameters, including the learning rate, for a wide range of baselines. Since our experimental setting follows this work, the hyperparameters, including the learning rate, are kept consistent with the code of this work. For the baselines that are not included in the code of DER but follow the experimental settings of DER, we follow their open-source code to set the hyperparameters, such as learning rate and epoch, which are also determined by grid-search. For the methods not following the experimental setting of DER, we conduct a grid-search on validation datasets to determine the hyperparameters, including the learning rate. When combining our LODE with other methods, we keep the learning rate consistent with the original methods.

Table 6: Variation of the first task's accuracy on Seq-CIFAR100.

| | After Task 1 | After Task 2 | After Task 3 | After Task 4 | After Task 5 |
|---|---|---|---|---|---|
| Remove $\ell_{ce}(\cdot; \mathcal{C}_n)$ | **87.40**$_{\pm 0.22}$ | 54.28$_{\pm 3.41}$ | 56.18$_{\pm 4.08}$ | 50.77$_{\pm 2.57}$ | 43.92$_{\pm 0.87}$ |
| Remove $\ell_n(\cdot)$ | **87.40**$_{\pm 0.22}$ | **82.38**$_{\pm 0.89}$ | **63.27**$_{\pm 0.75}$ | **54.18**$_{\pm 0.68}$ | **48.50**$_{\pm 0.95}$ |

Table 7: Results of combining with other methods.

| | Seq-CIFAR100 |
|---|---|
| PASS [47] | 47.45$_{\pm 0.37}$ |
| LODE (PASS) | **50.49**$_{\pm 0.22}$ |

## B.2 Experimental Setting in Online Continual Learning

**Datasets** We use three popular datasets for evaluation, including Seq-CIFAR10 [3], Seq-CIFAR100 [10], and Seq-TinyImageNet [22]. We follow existing online continual learning methods to partition tasks across different datasets for the purpose of evaluation. Specifically, Seq-CIFAR10 consists of 5 disjoint tasks with each task having 2 classes and 10k training samples. Seq-CIFAR100 consists of 20 disjoint tasks with each task having 5 classes and 2.5k training samples. Seq-TinyImageNet consists of 20 disjoint tasks with each task having 10 classes and 5k training samples. All the experiments are for the task-agnostic problem.

**Training Details** For all the datasets, we use reduced-ResNet18 as our model architecture, which has been commonly used in prior online continual learning methods [8, 41]. Reduced-ResNet18 is a variant of the original ResNet18 that has fewer channel sizes, and fewer filters overall. We use stochastic gradient descent (SGD) to update the model for all the methods. We set memory sizes M=5120 for all the datasets. Like the existing online continual learning works [8, 3], the replay size $|\mathcal{B}_{\mathcal{M}}|$ is the same as the mini-batch size $|\mathcal{B}_t|$ (fixed to 10). Data augmentation is the same as that in offline continual learning. Specifically, we use data augmentation such as random cropping and random rotation for both new mini-batch $\mathcal{B}_t$ and old mini-batch $\mathcal{B}_{\mathcal{M}}$.

## C  More Experimental Results

### C.1  Combine with More Methods

LODE can be applied in an exemplar-free setting. We apply our LODE to a popular exemplar-free continual learning method called prototype augmentation and self-supervision (PASS) [47], which is designed for task-agnostic problems and utilizes a cross-entropy loss for learning new tasks. Table 6 shows the variation of the accuracy for the first task during the learning of subsequent tasks. We can find that removing $\ell_n(\cdot)$ makes the model suffer from less forgetting than removing $\ell_{ce}(\cdot; \mathcal{C}_n)$. Table 7 shows the final results of PASS and LODE (PASS). We can find that LODE (PASS) outperforms PASS on Seq-CIFAR100.

### C.2  TSNE Visualization

We present the TSNE visualization in Figure 5, which shows the learned representations of different models on Seq-CIFAR10. As we can see, removing loss $\ell_n(\cdot)$ makes the representation of the old classes overlap less than removing loss $\ell_{ce}(\cdot; \mathcal{C}_n)$. Therefore, removing loss $\ell_n(\cdot)$ makes the model suffer from less forgetting than removing loss $\ell_{ce}(\cdot; \mathcal{C}_n)$. We will add this figure in the final version of our paper. Thanks for the suggestion.

### C.3  Plasticity of the Model

We also add a graph in Figure 6, which shows the accuracy of each new task during the training of Seq-CIFAR10. As we can see, removing either loss $\ell_{ce}(\cdot; \mathcal{C}_n)$ or loss $\ell_n(\cdot)$ decreases the model's plasticity and leads to lower performance on the new task. Hence, both of these two losses are necessary for the learning of a new task.

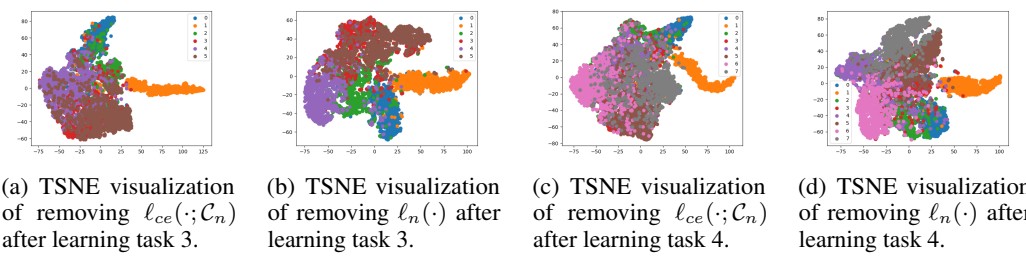

(a) TSNE visualization of removing $\ell_{ce}(\cdot; \mathcal{C}_n)$ after learning task 3.

(b) TSNE visualization of removing $\ell_n(\cdot)$ after learning task 3.

(c) TSNE visualization of removing $\ell_{ce}(\cdot; \mathcal{C}_n)$ after learning task 4.

(d) TSNE visualization of removing $\ell_n(\cdot)$ after learning task 4.

Figure 5: TSNE visualization

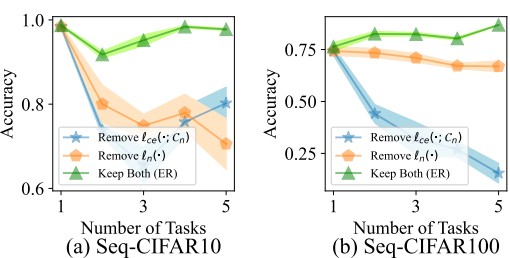

(a) Seq-CIFAR10

(b) Seq-CIFAR100

Figure 6: (a) and (b) show the variation of the new task's accuracy on different datasets.

## C.4 More Comparison

Table 8 presents the results of different methods on Seq-CIFAR100 with varying numbers of tasks. Here, the memory size is set as 500 for all the methods. As we can see, when the number of tasks varies, LODE still gives improvements on different methods. Furthermore, LODE (ESMER) still gets the best performance among all the methods.

Table 9 gives the results of additional methods that directly sample samples from $\mathcal{D}_t \cup \mathcal{M}$. Some methods like BIC and iCaRL, which directly sample samples from $\mathcal{D}_t \cup \mathcal{M}$, have been compared with our methods in the main text. Since all these methods keep an extra model in memory for distillation, we compare them with LODE (ESMER). As we can see, our method still gets the best performance.

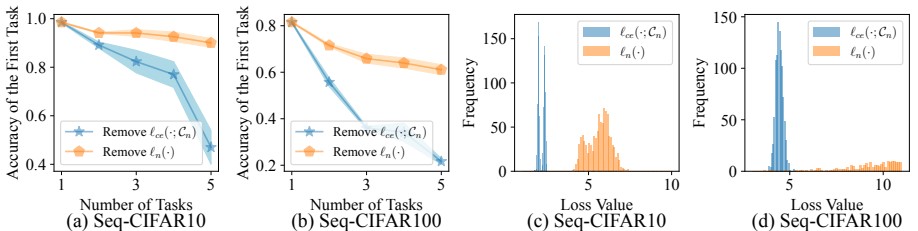

(a) Seq-CIFAR10

(b) Seq-CIFAR100

(c) Seq-CIFAR10

(d) Seq-CIFAR100

Figure 7: (a) and (b) show the variation of the first task's accuracy on different datasets. (c) and (d) show the distribution of different losses over different datasets before learning the second task. Here, the frequency corresponding to each loss value represents the number of samples with this loss value.

## C.5 More Analysis by Decoupling Loss

We also use dark experience replay++ (DER++) [7] to evaluate the impact of $\ell_{ce}(\cdot; \mathcal{C}_n)$ and $\ell_n(\cdot)$. Figure 7 gives the results. The results are similar to that in ER. Specifically, removing $\ell_n(\cdot)$ results in less forgetting of the first task than removing $\ell_{ce}(\cdot; \mathcal{C}_n)$. In other words, $\ell_n(\cdot)$ leads to more forgetting than $\ell_{ce}(\cdot; \mathcal{C}_n)$. Furthermore, the value of $\ell_n(\cdot)$ is larger than the value of $\ell_{ce}(\cdot; \mathcal{C}_n)$ on both Seq-CIFAR10 and Seq-CIFAR100.

Table 8: Results on Seq-CIFAR100 with different numbers of tasks. The memory size is set to 500.

| Keeping Extra Model | | $T = 4$ | $T = 10$ |
|---|---|---|---|
| no | PCR [25] | $30.96_{\pm0.56}$ | $19.88_{\pm2.29}$ |
| | MIR [3] | $30.44_{\pm0.65}$ | $22.01_{\pm0.68}$ |
| | ER-ACE [8] | $45.71_{\pm0.59}$ | $36.95_{\pm0.86}$ |
| | ER [9] | $30.72_{\pm0.27}$ | $20.45_{\pm1.38}$ |
| | LODE (ER) | $45.60_{\pm0.24}$ | $37.32_{\pm0.74}$ |
| | DER++ [7] | $43.68_{\pm0.86}$ | $36.30_{\pm0.85}$ |
| | LODE (DER++) | $\mathbf{48.95_{\pm0.77}}$ | $\mathbf{38.61_{\pm1.23}}$ |
| yes | CLS-ER [5] | $54.82_{\pm1.26}$ | $43.05_{\pm0.64}$ |
| | TAMiL [6] | $53.42_{\pm0.29}$ | $42.52_{\pm1.31}$ |
| | iCaRL [33] | $50.97_{\pm1.04}$ | $42.25_{\pm0.62}$ |
| | BIC [43] | $40.75_{\pm0.77}$ | $20.71_{\pm0.65}$ |
| | SSIL [1] | $39.39_{\pm0.36}$ | $33.06_{\pm0.86}$ |
| | ESMER [36] | $55.97_{\pm1.22}$ | $45.54_{\pm0.87}$ |
| | LODE (ESMER) | $\mathbf{57.92_{\pm0.30}}$ | $\mathbf{46.37_{\pm0.39}}$ |

Table 9: Results of more methods on Seq-CIFAR100 and Seq-TinyImageNet with memory size 500.

| | Seq-CIFAR100 | Seq-TinyImageNet |
|---|---|---|
| iCaRL [33] | $49.59_{\pm0.95}$ | $20.01_{\pm0.50}$ |
| BIC [43] | $37.06_{\pm0.60}$ | $29.82_{\pm0.88}$ |
| SSIL [1] | $41.61_{\pm0.37}$ | $16.80_{\pm0.71}$ |
| PODNet [14] | $45.95_{\pm0.45}$ | $26.10_{\pm1.71}$ |
| UCIR [17] | $47.56_{\pm0.43}$ | $26.22_{\pm0.84}$ |
| LODE (ESMER) | $\mathbf{55.06_{\pm0.35}}$ | $\mathbf{32.15_{\pm0.17}}$ |

