# OpenReview forum: "Loss Decoupling for Task-Agnostic Continual Learning"
_NeurIPS.cc/2023/Conference — NeurIPS 2023 poster_

### Official Review · Reviewer_Tdag · 2023-07-05

**Soundness:** 3 good
**Presentation:** 3 good
**Contribution:** 3 good
**Rating:** 6
**Confidence:** 3

**Summary:**

This paper investigates the stability-plasticity tradeoff problem in a task-agnostic continual learning (CL) setting by decoupling the loss of the new task (LODE). LODE introduces two separate objectives for the new task: new/old class distinction and new class distinction. The authors analyze the impact of these objectives on forgetting and propose a strategy that assigns different weights to achieve a better stability-plasticity tradeoff. The proposed method is extensively evaluated through experiments and ablation studies on CIFAR10, CIFAR100, and TinyImageNet, demonstrating its effectiveness.

**Strengths:**

- The proposed idea is simple and interesting, offering a promising approach to improve the stability-plasticity tradeoff. It is likely to gain attention within the CL community.

- The submission is well-written and easy to follow.

- The experimental settings and comparisons are detailed, considering both offline and online settings.

**Weaknesses:**

- The experiments only employ ResNet18; it would be beneficial to include a larger model for evaluation.

- The paper lacks a description of the learning rate. Is it the same for all compared methods? Was a grid-search conducted to determine it?

- The use of TSNE visualization could provide a better way to validate the assumptions made.

**Questions:**

- Can LODE be applied in an exemplar-free setting? It would be helpful to have a discussion or a simple experiment addressing this.

- The proposed method is only combined with similar replay methods (ER, DER, and ESMER). More combinations or a discussion of this limitation would be valuable.

**Limitations:**

Yes.

---

> ### Author Rebuttal · Authors · 2023-08-10
>
> **Q1: The experiments only employ ResNet18; it would be beneficial to include a larger model for evaluation.**
>
> **A1**: Our experiments are built on top of the mammoth [3] continual learning repository in PyTorch like many existing works [2,3]. This repository consists of only ResNet18 and a small MLP for all the experiments. To make our experiments convincing, we do not change the model architecture. CL with larger models is also very interesting, but this is not the focus of this paper. We leave it for future study. Thanks for the suggestion.
>
> **Q2: The paper lacks a description of the learning rate. Is it the same for all compared methods? Was a grid-search conducted to determine it?**
>
> **A2**: Yes, the learning rate is determined by a grid-search. Specifically, the open-source code of the existing work DER has conducted a grid-search to determine hyperparameters, including the learning rate, for a wide range of baselines. Since our experimental setting follows this work, the hyperparameters, including the learning rate, are kept consistent with the code of this work. For the baselines that are not included in the code of DER but follow the experimental settings of DER, we follow their open-source code to set the hyperparameters, including the learning rate, which is also determined by a grid-search. For the methods not following the experimental setting of DER, we conduct a grid-search on validation datasets to determine the hyperparameters, including the learning rate. When combining our LODE with other methods, we keep the learning rate consistent with the original methods. We will add the description of the learning rate in the final version of our paper. Thanks for the suggestion.
>
> **Q3: The use of TSNE visualization could provide a better way to validate the assumptions made.**
>
> **A3**: We present the TSNE visualization in Figure III of the uploaded PDF. As we can see, removing loss $l_n(\cdot)$ makes the representation of the old classes overlap less than removing loss $l_{ce}(\cdot;C_n)$. Therefore, removing loss $l_n(\cdot)$ makes the model suffer from less forgetting than removing loss $l_{ce}(\cdot;C_n)$. We will add this figure in the final version of our paper. Thanks for the suggestion.
>
> **Q4: Can LODE be applied in an exemplar-free setting? It would be helpful to have a discussion or a simple experiment addressing this.**
>
> **A4**: YYes, LODE can be applied in an exemplar-free setting. We apply our LODE to a popular exemplar-free continual learning method called prototype augmentation and self-supervision (PASS) [1], which is designed for task-agnostic problems and utilizes a cross-entropy loss for learning new tasks. Specifically, we decouple the loss in PASS in the same way as Equation (3) and apply our LODE to it. The following table shows the results of PASS and LODE (PASS). We can find that LODE (PASS) outperforms PASS on Seq-CIFAR100. More results of combining our LODE with exemplar-free methods will be added in the final version of our paper. Thanks for the suggestion.
>
> | Methods       |  ACC (%)
> |:-|:-:
> | PASS          | 47.45±0.37
> | LODE (PASS)  | **50.49±0.22**
>
> **Q5: The proposed method is only combined with similar replay methods (ER, DER, and ESMER). More combinations or a discussion of this limitation would be valuable.**
>
> **A5**: From the methodology perspective, any methods with the loss in the form of Equation (1) can be combined with our methods. We combined with ER, DER, and ESMER since these methods are either state-of-the-art methods or competitive in a wide range of settings, including both offline and online continual learning setting. In **A4**, we present the combination of our method with other continual learning method. We will add the results and further discussion of limitations of the experiments in the final version of our paper. Thanks for the suggestion.
>
> [1] Zhu F, Zhang X Y, Wang C, et al. Prototype augmentation and self-supervision for incremental learning. CVPR, 2021.
>
> [2] Sarfraz F, Arani E, Zonooz B. Error Sensitivity Modulation based Experience Replay: Mitigating Abrupt Representation Drift in Continual Learning. ICLR, 2023.
>
> [3] Buzzega P, Boschini M, Porrello A, et al. Dark experience for general continual learning: a strong, simple baseline. NeurIPS, 2020.

---

> > ### Comment · Reviewer_Tdag · 2023-08-17
> >
> > Thank you for the authors' efforts in the rebuttal. Their response has addressed my concerns, and I will maintain my current rating.

---

> > > ### Author Response · Authors · 2023-08-20
> > >
> > > Thanks a lot for your response. We would like to inquire if you have any other concerns that might require further elaboration. We would be happy to provide additional explanations  as needed.

---

### Official Review · Reviewer_n7mC · 2023-07-06

**Soundness:** 3 good
**Presentation:** 2 fair
**Contribution:** 2 fair
**Rating:** 5
**Confidence:** 4

**Summary:**

The paper titled "Loss Decoupling for Continual Learning" addresses the challenge of catastrophic forgetting in continual learning, where a model needs to learn multiple tasks sequentially. The paper focuses on the task-agnostic problem, where task identities are not available during inference. The main contribution of the paper is the proposal of a method called loss decoupling (LODE) for continual learning. LODE separates the learning objectives for a new task by decoupling the loss, allowing for different weights to be assigned to different objectives. Experimental evaluations on multiple datasets demonstrate that LODE outperforms existing state-of-the-art replay-based methods, achieving a better trade-off between stability and plasticity.

**Strengths:**

1. The paper introduces a novel method, LODE, which decouples the learning objectives for a new task in continual learning. This approach offers a fresh perspective on addressing the trade-off between stability and plasticity. The idea of assigning different weights to different objectives based on their impact on forgetting is unique and contributes to the advancement of continual learning methods.
2. The paper demonstrates a high level of quality in terms of experimental design, clear presentation of results, and thorough analysis. The use of popular datasets, the comparison with state-of-the-art methods, and the inclusion of multiple runs with different seeds enhance the reliability and validity of the experimental findings. The clarity of the paper's structure and explanations contributes to its overall quality.
3. The experimental results demonstrate the effectiveness of LODE in outperforming existing state-of-the-art methods, which has practical implications for enhancing continual learning algorithms and facilitating real-world applications.

**Weaknesses:**

1. The paper focuses solely on the task-agnostic problem and does not explore the separation of learning objectives in other continual learning problems. It would be valuable for future work to investigate the application of loss decoupling to different problem settings, thereby broadening the scope and applicability of the proposed method.
2. Lack of comparison with other loss decoupling methods [1][2]: While the paper proposes LODE as a novel method, it does not compare LODE with other loss decoupling techniques, if any exist. Including such a comparison would strengthen the paper's novelty and showcase the advantages of LODE over alternative approaches.
3. The experiments primarily consider image classification tasks, and the paper does not thoroughly explore the effectiveness of LODE for other types of tasks. Conducting experiments and providing insights on the generalizability of LODE to different domains would enhance the paper's impact and practical relevance.
4. The application of the proposed method in continual learning has strong limitations: 1) It must be a replay-based method and satisfy the loss Equation 1; 2) It must be a task-agnostic setting. However, the paper uses a very broad title, covering the entire field of continual learning.

[1] Sun, W., Li, Q., Zhang, J., Wang, W., & Geng, Y. A. (2023). Decoupling Learning and Remembering: A Bilevel Memory Framework With Knowledge Projection for Task-Incremental Learning. In Proceedings of the IEEE/CVF Conference on Computer Vision and Pattern Recognition (pp. 20186-20195).
[2] Zhao, Z., Zhang, Z., Tan, X., Liu, J., Qu, Y., Xie, Y., & Ma, L. (2023). Rethinking Gradient Projection Continual Learning: Stability/Plasticity Feature Space Decoupling. In Proceedings of the IEEE/CVF Conference on Computer Vision and Pattern Recognition (pp. 3718-3727).

**Questions:**

1. Could you elaborate on how the proposed LODE method can be applied to other continual learning problems beyond the task-agnostic setting? What are the potential challenges and considerations in extending LODE to different problem formulations?
2. Can you provide further insights or examples regarding the generalizability of LODE to other types of tasks, apart from image classification?
3. Regarding the limitation of focusing on the task-agnostic problem, have you identified any potential strategies or directions for separating learning objectives in other types of continual learning problems? If not, what are the challenges or considerations in extending loss decoupling to task-aware problems?
4. How do you get this Figure1 (b)? Is this based on some specific data set or an ideal description? What does the y-axis represent?

**Limitations:**

/

---

> ### Author Rebuttal · Authors · 2023-08-10
>
> **Q1: The paper focuses solely on the task-agnostic problem. ... investigate the application of loss decoupling to different problem settings.**
>
> **A1**: Task-agnostic problem we consider in this work is a challenging problem where task identities are not available during testing. In contrast, for another popular problem in continual learning (CL), called task-aware problem, the model can obtain task identities during testing. Most CL methods, including the work that the reviewer mentioned, consider only one of them. Specifically, the work mentioned by the reviewer, called bilevel memory system with knowledge projection (BMKP) [1], considers only task-aware problem, as shown in Section 3.2 of their paper and their code. In contrast, our method considers task-agnostic problem, which is more challenging than task-aware problem. In **A6**, we give the insight about broadening the applicability of our proposed method to different problem settings, which will be our future work. Thanks for the suggestion.
>
> **Q2: Lack of comparison with other loss decoupling methods [1][2].**
>
> **A2**: We do not compare with the methods that the reviewer mentioned since these methods consider task-aware problem while our method considers task-agnostic problem. Specifically, BMKP [1] requires the task identities to choose the corresponding knowledge representations during testing, making it unsuitable in the task-agnostic problem. Space decoupling (SD) [2] does not explicitly mention that it only considers task-aware problem, but its experiments completely follow a task-aware CL method called TRGP [3], indicating its focus on task-aware problem. Furthermore, both these methods are published at CVPR 2023. Since the conference data of CVPR 2023 was after NeurIPS 2023 submission deadline, we were not aware of these two methods when we submitted our paper. We will cite these two methods in the final version of our paper and discuss the difference between our method and them.
>
> **Q3: The experiments primarily consider image classification tasks, ... not explore other types of tasks.**
>
> **A3**: We primarily consider image classification tasks since many existing methods, including the works the reviewer mentioned [1][2], consider primarily image classification tasks. We have introduced this limitation in Section 6 of our paper, which will be further studied in future work.
>
> **Q4: limitations: 1 It must be a replay-based method and satisfy Equation 1; 2 It must be a task-agnostic setting. the paper uses a very broad title.**
>
> **A4**: First, replay-based methods are among the most representative CL methods, and most existing state-of-the-art replay-based methods satisfy Equation (1). Second, task-agnostic problem is challenging and many existing replay-based methods focus on this problem. We will change the title to “loss decoupling for task-agnostic continual learning” in the final version of our paper. Thanks for the suggestion.
>
> **Q5: provide further insights regarding the generalizability of LODE to other types of tasks?**
>
> **A5**: First, we should emphasize that many CL methods, including BMKP and SD mentioned by the reviewer, focus solely on image classification. Second, from the methodology perspective, our methods can be applied to other tasks like NLP and speech data analysis in the task-agnostic problem, where the learning of a new task can be decoupled into learning a new task itself and learning to distinguish between the new task and the old tasks. This is because the main contribution of our method lies in decoupling the loss function which is a general one for modeling different types of input data.
>
> **Q6: elaborate how LODE method can be applied to other continual learning problems? identify potential strategies for separating learning objectives in other continual learning problems?**
>
> **A6**: We can leverage the semantic information of the classes to separate learning objectives in other types of CL problems including task-aware problems. Taking the task-aware problem as an example, assume the model is going to learn to distinguish three classes, including tiger, wolf and bee. Assume the model has learned to distinguish between cat and dog. Since tiger is similar to cat and wolf is similar to dog semantically, the model may change very little when learning to distinguish between tiger and wolf. Note that changing less is conducive to overcome catastrophic forgetting, we can divide the new task into two learning objectives, where the first objective makes the model distinguish between tiger and wolf, and the second objective makes the model distinguish between other category pairs. We will add this discussion in the final version of our paper. Thanks for the suggestion.
>
> **Q7: How do you get Figure 1 (b)? based on some specific data set or an ideal description? What does the y-axis represent?**
>
> **A7**: Figure 1 (b) is an idea description. This figure describes the coupling property of existing replay-based methods. Specifically, existing replay-based methods either sample from $\mathcal{D}_t$ and $\mathcal{M}$ separately, or directly sample from $\mathcal{D}_t⋃\mathcal{M}$. Then, they compute the loss for the new task through cross-entropy, which couples different new learning objectives. Figure 1 (b) is used to introduce this phenomenon figuratively. The y-axis represents the model’s abilities, including its ability to adapt itself to the new task (plasticity) and its ability to overcome forgetting (stability). This explanation will be added in the final version. Thanks for the comment.
>
> [1] Sun W, Li Q, Zhang J, et al. Decoupling Learning and Remembering: A Bilevel Memory Framework with Knowledge Projection for Task-Incremental Learning. CVPR, 2023.
>
> [2] Kim S, Noci L, Orvieto A, et al. Achieving a Better Stability-Plasticity Trade-off via Auxiliary Networks in Continual Learning. CVPR, 2023.
>
> [3] Lin S, Yang L, Fan D, et al. TRGP: Trust Region Gradient Projection for Continual Learning. ICLR, 2022.

---

> > ### Comment · Reviewer_n7mC · 2023-08-18
> > **Response to the authors**
> >
> > I thank the authors for the detailed response, which addresses most of my concerns. I update my rating to Borderline accept.

---

> > > ### Author Response · Authors · 2023-08-20
> > >
> > > Thanks a lot for your response. We would like to inquire if you have any other concerns that might require further elaboration. We would be happy to provide additional explanations as needed.

---

### Official Review · Reviewer_odRp · 2023-07-06

**Soundness:** 3 good
**Presentation:** 3 good
**Contribution:** 3 good
**Rating:** 7
**Confidence:** 3

**Summary:**

Several replay based continual learning methods decouple their loss into a loss based on samples from the replay to maintain performance on old tasks, and a loss based on data from the current task to learn the new task. This paper proposes to further decouple the latter loss into a component that helps the model distinguish whether a sample is in the new task or from previous tasks, and a component helps the model distinguish between just the new classes in the current task. The paper shows that decoupling this loss and giving different weight to the two components helps in maintaining performance and results in better final accuracy for Seq-CIFAR10, Seq-CIFAR100, and Seq-TinyImagenet.

**Strengths:**

- The paper proposes an intuitive decoupling of the loss used in many continual learning works. It shows that doing this decoupling is useful, and can improve performance by weighting the different parts of the decoupled loss differently.
- The paper’s experiments and presentation of experiments are well done, although I do ask the authors to address some of the questions I present below. The hyperparameter sensitivity analysis is useful in showing that the method is at least somewhat robust to hyperparameter changes.

**Weaknesses:**

The main weaknesses are mostly related to the additional questions and experiments I ask for in the next section. There are a few experiments I’d like to see that could make the paper stronger.

**Questions:**

- For section 4.2.3, did you do a hyperparameter search for the values of $\beta_1$ and $\beta_2$ for the methods that weren’t yours?
- The $l_n$ loss in the denominator is essentially a binary classification loss. Since you are only using positive examples to compute this loss, it increases the value of the logits of the current task and decreases the value of the logits of previous tasks for your current data. Did you try also using some data from the replay buffer for this loss?
- For Figure 2, it might be helpful to also see on the graph a line where neither component is removed (ie both components of the decoupled loss without reweighting), and a line with the coupled loss used in traditional ER (ie Equation 2). This could tell us how much the decoupling formulation that is proposed actually corresponds to the usual coupled formulation. Another graph to add could be the accuracy on the new task being trained, allowing us to get a sense of how these losses affect plasticity.

**Limitations:**

The work presents a basic discussion of limitations and societal impacts. The limitations section could be further developed.

---

> ### Author Rebuttal · Authors · 2023-08-10
>
> **Q1: For section 4.2.3, did you do a hyperparameter search for the values of $β_1$ and $β_2$ for the methods that weren’t yours?**
>
> **A1**: In Section 4.2.3, we did not do a hyperparameter search for the values of $β_1$ and $β_2$ for the methods that weren’t ours and kept their value consistent with our method. To be more rigorous, we add the hyperparameter search for the values of $β_1$ and $β_2$ for the methods that weren’t ours and present them in the following table. The results lead to the same conclusion as that in Section 4.2.3. We will replace Table 2 with this table in the final version of our paper. Thanks for the suggestion.
>
> |  LODE (DER++)|Seq-CIFAR10|Seq-CIFAR100
> |:-|:-:|:-:
> | $β_1=C,β_2=ρ\frac{\vert C_n\vert}{\vert C_o\vert}$  | **75.45±0.90**   | **46.31±1.01**
> | $β_1=β_2=ρ\frac{\vert C_n\vert}{\vert C_o\vert}$    | 71.18±0.80   | 37.49±1.79
> | $β_1=β_2=C$                   | 73.80±0.20   | 42.08±1.71
> | $β_1=ρ\frac{\vert C_n\vert}{\vert C_o\vert},β_2=C$  | 73.19±0.15   | 40.79±0.12
> |  **LODE (ESMER)**   | **Seq-CIFAR10** | **Seq-CIFAR100**|
> | $β_1=C,β_2=ρ\frac{\vert C_n\vert}{\vert C_o\vert}$  | **74.53±0.95**   | **55.06±0.35**  |
> | $β_1=β_2=ρ\frac{\vert C_n\vert}{\vert C_o\vert}$    | 73.41±0.40   |  45.64±0.87 |
> | $β_1=β_2=C$                   | 73.08±0.81   | 52.37±0.87  |
> | $β_1=ρ\frac{\vert C_n\vert}{\vert C_o\vert},β_2=C$  | 72.38±0.24   | 51.86±0.35  |
>
> **Q2: The loss $l_n$ in the denominator is essentially a binary classification loss. Since you are only using positive examples to compute this loss, it increases the value of the logits of the current task and decreases the value of the logits of previous tasks for your current data. Did you try also using some data from the replay buffer for this loss?**
>
> **A2**: We did not try using any data from replay buffer for this loss since the replay loss $l_{rep}(\cdot)$ has already included the cross-entropy loss $l_{ce}(\cdot)$ computed by the samples of old classes in most replay-based methods. At this time, we can decouple the loss $l_{ce}(\cdot)$ included in $l_{rep}(\cdot)$ in a similar way as described in Equation (3) of the paper:
> $$l_{ce}(f_Θ(x),y)=-log⁡(\frac{\exp⁡(o_y)}{∑_{i=1}^{m+n}\exp⁡(o_i)})-log⁡(\frac{∑_{i=1}^m \exp⁡(o_i)}{∑_{i=1}^{m+n}\exp⁡(o_i)}).$$
> Here, $m$ and $n$ denote the number of old and new classes, respectively. The second term of this equation is the loss $l_n(\cdot)$ computed using the negative examples. Since the replay loss $l_{rep}(\cdot)$ has already included the loss $l_n(\cdot)$ computed using the negative examples, we did not try using some data from the replay buffer for this loss.
>
> **Q3: For Figure 2, it might be helpful to also see on the graph a line where neither component is removed (ie both components of the decoupled loss without reweighting), and a line with the coupled loss used in traditional ER (ie Equation 2). This could tell us how much the decoupling formulation that is proposed actually corresponds to the usual coupled formulation. Another graph to add could be the accuracy on the new task being trained, allowing us to get a sense of how these losses affect plasticity.**
>
> **A3**: We add the line where neither component is removed in Figure I of the uploaded PDF. As we can see, when neither component is removed from Equation (2), the model forgets either more than or similarly to removing loss $l_{ce}(\cdot;C_n)$. We also add a graph in Figure II of the uploaded PDF, which shows the accuracy of each new task during the training. As we can see, removing either loss $l_{ce}(\cdot;C_n)$ or loss $l_n(\cdot)$ decreases the model’s plasticity and leads to lower performance on the new task. Hence, both of these two losses are necessary for the learning of a new task. We will add these graphs in the final version of our paper. Thanks for the suggestion.
>
> [1] Buzzega P, Boschini M, Porrello A, et al. Dark experience for general continual learning: a strong, simple baseline. NeurIPS, 2020.

---

> > ### Comment · Reviewer_odRp · 2023-08-16
> >
> > I am satisfied with the response, and have raised my score accordingly.

---

> > > ### Author Response · Authors · 2023-08-20
> > >
> > > Thanks a lot for your response. We would like to inquire if you have any other concerns that might require further elaboration. We would be happy to provide additional explanations as needed.

---

### Official Review · Reviewer_Y5HQ · 2023-07-07

**Soundness:** 3 good
**Presentation:** 2 fair
**Contribution:** 2 fair
**Rating:** 5
**Confidence:** 3

**Summary:**

This paper investigated class incremental continual learning (CL). By showing that two objectives, i.e., classifying among new classes and classifying between new classes and old classes, can have different impacts on the performance of CL, this paper proposed to decouple the new task loss and assign different weights to these two objectives. Experiments are conducted on multiple datasets to verify the effectiveness of the proposed method LODE in comparison with related baseline methods.

**Strengths:**

It is interesting to investigate the different impacts inter-task and intra-task classification on the performance of class incremental CL.

**Weaknesses:**

1. The results in Figure 2(a) seems very counter-intuitive. When removing $l_n$, the training of the current task only focuses on the classification within the task and ignores the differentiation from old classes in the memory. In this case, the forgetting should be larger compared to the case of removing $l_{ce}$, while in Figure 2(a) the forgetting was smaller. The authors didn't give a convincing explanation of the underlying reason for the phenomenon.

 2. The weight selection (5) in LODE is entirely based on heuristics and manually determined. It is hard to justify the selection of the weights, but the performance of LODE is highly sensitive to the weight selection as shown in Table 2.

**Questions:**

1. How did you select the values of $C$ and $\rho$ in (5)?

2. Conceptually, the loss decoupling can be used in CL without using data replay. In this way, wouldn't it be more clear to understand the impact of these two parts on the performance of CL?

**Limitations:**

Yes

---

> ### Author Rebuttal · Authors · 2023-08-10
>
> **Q1: The results in Figure 2(a) seem very counter-intuitive. The authors didn't give a convincing explanation of the underlying reason for the phenomenon.**
>
> **A1**: The results in Figure 2 (a) of the paper are intuitively reasonable. First, since replay-based methods usually keep limited samples in memory, when the model learns a new task, it has access to much fewer samples from the old classes than from the new classes. Therefore, utilizing loss $l_n(\cdot)$ to learn to distinguish between new classes and old classes introduces a risk of biasing the model towards the new classes, potentially leading to serious catastrophic forgetting [1]. In contrast, loss $l_{ce}(\cdot;C_n)$ is independent of the old classes, thereby avoiding introducing a risk of biasing the model towards the new classes. Second, as shown in Figure 2 (c) of the paper, the value of loss $l_n(\cdot)$ is typically larger than the value of $l_{ce}(\cdot;C_n)$ during training. This finding is consistent with existing work [3], which suggests that larger loss for the new task may result in larger feature drift, leading to more forgetting. Based on these two reasons, it is intuitively reasonable that loss $l_n(\cdot)$ causes larger forgetting for the old classes than loss $l_{ce}(\cdot;C_n)$, implying that removing loss $l_n(\cdot)$ results in less forgetting than removing loss $l_{ce}(\cdot;C_n)$. Thanks for the information. We will add this explanation in the final version of our paper.
>
>
> **Q2: The weight selection (5) in LODE is entirely based on heuristics and manually determined. It is hard to justify the selection of the weights, but the performance of LODE is highly sensitive to the weight selection as shown in Table 2.**
>
> **A2**: Actually, the weight selection (5) in LODE is not based on heuristics and manually determined. We give the explanation for Equation (5) in Section 3.3. First, loss $l_n(\cdot)$ aims to distinguish new classes from the old classes and the replay loss $l_{rep}(\cdot)$ in Equation (4) keeps the performance on the old classes. Minimizing losses $l_n(\cdot)$ and $l_{rep}(\cdot)$ together makes the model learn to distinguish whether a given sample belongs to the new classes or the old classes, which is a binary classification task. If we set both the weights of $l_n(\cdot)$ and $l_{rep}(\cdot)$ as constants during the whole learning process, the model may suffer from large forgetting for the old classes. More specifically, when the number of tasks is large and each new task only increases a small number of new classes, the number of old classes is much larger than the number of new classes. At this time, setting the weight of $l_n(\cdot)$ as a constant may introduce a risk of biasing the model towards the new classes. On the contrary, setting the weight of $l_n(\cdot)$ proportional to the ratio $\frac{|C_n|}{|C_o|}$ removes this bias since the weight of $l_n(\cdot)$ will be small when the number of old classes is much larger than the number of new classes. Different from $l_n(\cdot)$, loss $l_{ce}(\cdot;C_n)$ is independent of the old classes and thus it does not introduce a risk of biasing the model towards the new classes. Therefore, we set the weight of $l_{ce}(\cdot;C_n)$ as a constant C to maintain the model’s plasticity. Table 2 verifies the effectiveness of this weight selection strategy.
>
> **Q3: How did you select the values of C and ρ in (5)?**
>
> **A3**: We keep the value of C as 1 for all the experiments in Table 1 and Table 3, which we find is enough for the model to perform well. We follow existing continual learning method DER [4] to select the value of ρ. Specifically, we select the value of ρ through grid-search on a validation set obtained by sampling 5% of the training set. We investigate the influence of these two hyperparameters in Figure 4.
>
> **Q4: Conceptually, the loss decoupling can be used in CL without using data replay. In this way, wouldn't it be more clear to understand the impact of these two parts on the performance of CL?**
>
> **A4**: Thanks a lot for this informative comment. Yes, the loss decoupling can be used in some CL methods without using data replay as long as these methods use a standard cross-entropy loss to learn new tasks, which is the common choice for many CL methods. We focus on replay-based methods since they are more effective for task-agnostic problem due to the replaying of old data, particularly in the online continual learning setting. Here, we choose a CL method without using data replay, called prototype augmentation and self-supervision (PASS) [2] to evaluate our methods. The following two tables show the results of PASS on Seq-CIFAR100. The results are consistent with CL methods using data replay. Specifically, the first table shows the variation of the accuracy for the first task during the learning of subsequent tasks. We can find that removing $l_n(\cdot)$ makes the model suffer from less forgetting than removing $l_{ce}(\cdot;C_n)$. Table 2 shows that LODE (PASS) outperforms PASS. We will add these results in the final version of our paper.
>
> ||After Task 1|After Task 2|After Task 3|After Task 4|After Task 5
> |:-|:-:|:-:|:-:|:-:|:-:
> |Remove $l_{ce}(\cdot;C_n)$|**87.40±0.22**|54.28±3.41|56.18±4.08|50.77±2.57|43.92±0.87
> |Remove $l_n(\cdot)$|**87.40±0.22**|**82.38±0.89**|**63.27±0.75**|**54.18±0.68**|**48.50±0.95**
>
> ||ACC (%)
> |:-|:-:
> |PASS|47.45±0.37
> |LODE (PASS)|**50.49±0.22**
>
> [1] Ahn H, Kwak J, Lim S, et al. Ss-il: Separated softmax for incremental learning. ICCV, 2021.
>
> [2] Zhu F, Zhang X Y, Wang C, et al. Prototype augmentation and self-supervision for incremental learning. CVPR, 2021.
>
> [3] Sarfraz F, Arani E, Zonooz B. Error Sensitivity Modulation based Experience Replay: Mitigating Abrupt Representation Drift in Continual Learning. ICLR, 2023.
>
> [4] Buzzega P, Boschini M, Porrello A, et al. Dark experience for general continual learning: a strong, simple baseline. NeurIPS, 2020.

---

> > ### Comment · Reviewer_Y5HQ · 2023-08-18
> >
> > Thanks a lot for the response. It clarifies most of my questions. I will raise my score to 5. While the current weight selection strategy sounds reasonable, I do think that a more principled way, e.g., through optimization, could further improve the quality of this paper.

---

> > > ### Author Response · Authors · 2023-08-20
> > >
> > > Thanks a lot for your response. Previously, we performed weight selection through optimization. Specifically, we treated the weights of different losses as learnable parameters and formulated the weight selection as a bilevel optimization problem. However, our findings showed that this approach did not give notable improvements compared to using Equation (5) for weight selection. Additionally, it introduced extra computational overhead. If the reviewer could provide a specific reference, method, or even a code related to this, we would greatly appreciate it.

---

### Author Rebuttal · Authors · 2023-08-10

We appreciate the reviewers' positive comments:
1. The problem is interesting to investigate the different impacts inter-task and intra-task classification on the performance of class incremental CL. (reviewer Y5HQ)
2. The proposed method is intuitive and can improve performance in many continual learning works. / The proposed method contributes to the advancement of continual learning methods. (reviewer odRp, reviewer n7mC)
3. This paper offers a promising approach to improve the stability-plasticity tradeoff. / The paper introduces a novel method and offers a fresh perspective on addressing the trade-off between stability and plasticity. (reviewer Tdag, reviewer n7mC)
4. The submission is well-written and easy to follow. / The clarity of the paper's structure and explanations contributes to its overall quality. (reviewer Tdag, reviewer n7mC)
5. The experimental design demonstrates a high level of quality. / The paper’s experiments and presentation of experiments are well done. / The experimental settings and comparisons are detailed. (reviewer n7mC, reviewer odRp, reviewer Tdag)

At the same time, we are grateful for the valuable questions and concerns of all the reviewers and have provided individual responses accordingly.

---

### Decision · Program_Chairs · 2023-09-21

**Decision:**

Accept (poster)

**Comment:**

The rebuttal and discussion convinced three reviewers to raise their score leading to an overall positive recommendation from the reviewers (5567). The AC follows the recommendation of the reviewers and recommends acceptance. Given the importance of the rebuttal/discussion, the authors should carefully incorporate these into their final version, including the improved explanations, improved analysis hyperparameters, and inclusion of missing related work.